# The Impact of Clinical Atropine Use in Taiwanese Schoolchildren: Changes in Physiological Characteristics and Visual Functions

**DOI:** 10.3390/children8111054

**Published:** 2021-11-15

**Authors:** Hui-Ying Kuo, Ching-Hsiu Ke, Shyan-Tarng Chen, Han-Yin Sun

**Affiliations:** 1Department of Optometry, Chung Shan Medical University, Taichung 402, Taiwan; evekuo@csmu.edu.tw (H.-Y.K.); shyan@csmu.edu.tw (S.-T.C.); 2Department of Ophthalmology, Chung Shan Medical University Hospital, Taichung 402, Taiwan; 3Department of Optometry, Central Taiwan University of Science and Technology, Taichung 406, Taiwan; 108364@ctust.edu.tw

**Keywords:** myopia control, atropine, the amplitude of accommodation, pupil size, intraocular pressure, visual complaint

## Abstract

Taiwan is commonly noted for its high prevalence of myopia, as well as a long history of more than 20 years of using atropine to control myopia. However, the clinical implications are rarely discussed. This is a cross-sectional study investigating the influence of topical atropine instillation on ocular physiology, visual function, and visual discomfort in children. Aged 7 to 12 years, 212 schoolchildren were recruited and divided into the atropine group and the non-atropine group. Physiological characteristics such as pupil size and intraocular pressure were measured, and a variety of visual functions was also evaluated. A questionnaire was used to investigate the side effects and visual complaints caused by atropine treatment. There was a significant difference in pupil size (OD: 5.40 ± 0.90 vs. 6.60 ± 1.01 mm; OS: 5.42 ± 0.87 vs. 6.64 ± 1.00 mm, *p* < 0.001) between the two groups. Reductions in near visual acuity, accommodation, convergence ability, and stereopsis were observed in the atropine group. The horizontal pupil diameter enlarged, and visual functions were greatly affected after administration of topical atropine. The changes in visual function during atropine therapy need to be carefully monitored by clinicians, while patient compliance is usually the key to success.

## 1. Introduction

Myopia and high myopia influenced 27% (1893 million) and 2.8% (170 million) of the world population in 2015, respectively. It is also alarming that over half of the world’s population will be affected by myopia by 2050, particularly in the younger population [1]. The epidemic level of myopia (greater than 80% of the whole population) causes a huge impact in young age groups, particularly in urban regions. In some Asian countries such as Korea, Taiwan, and Singapore, the prevalence is between 84% and 97% [2,3,4]. It was also evident that there is a rising trend in the severity of myopia in the United States [5] as well as worldwide [6]. High myopia brings further vision challenges as it increases the risk of developing sight-threatening pathological conditions, such as glaucoma, retinal detachment, chorio-retinal atrophy, and lacquer cracks, which may result in irreversible visual impairments^6^. Recent studies reported that more than 15% of Asian high school students had high myopia (>6 diopters), which increases the risk of future pathological consequences of myopia [7,8]. The detection and treatment of myopic pathogenesis complications should be warranted in the population of high myopia, and is required to prevent the onset or progression of myopia in younger generations.

Some control methods, including optical corrections such as bifocal spectacle lenses, ortho-keratologic lenses, multifocal contact lenses, increased exposure to outdoor activities and the use of atropine, have been shown to slow down myopia progression with differential effectiveness as well as side effects [9,10]. Atropine has been regarded as the most commonly used and studied pharmacological agent for myopia control, even though the exact mechanism of topical atropine is still unknown. The chick model has proved that the administration of the muscarinic antagonist atropine could prevent experimentally induced myopia through a non-accommodative pathway [11]. Atropine is a non-selective antimuscarinic agent that shows a high affinity to M1-M5 receptors and thus results in mydriasis and cycloplegia in the pupillary sphincter ciliary muscle. After receiving 1% topical atropine, the treated mice’s eyes showed a reduction in their lens thickness and vitreous chamber depth. The message levels for M1, M3, and M4 were upregulated in myopic sclera after atropine treatment, but M2 and M5 showed little change when using the real-time polymerase chain reaction technique [12]. The up-and-down regulation of retinal and scleral muscarinic receptors with influence on the sclera matrix has been postulated [13,14]. New theories were also proposed that extra-ultraviolet A exposure caused by pupillary dilation may limit axial elongation [15] or that myopia development might be associated with choroidal thickness in animal models after intravitreal injection of atropine and slowed axial elongation [16].

Since the early trials in the 1970s [17,18], numerous studies have demonstrated the effect of atropine in slowing down myopia progression and is currently the most extensively used pharmaceutical agent for myopia control. Several retrospective studies have shown that high-dose atropine (0.5–1%) is the most effective in slowing the progression of myopia, with its control efficacy up to almost 80% [19,20,21]. In 1989, Yen et al. reported that 1% atropine group (−0.22 ± 0.54 D per year) had a better control effect on myopic progression when compared to 1% cyclopentolate (−0.58 ± 0.49 D per year) and placebo (−0.91 ± 0.58 D per year) groups after one-year follow-up [19]. Shin et al. compared the subjects who used tropicamide with those who had 0.5, 0.25, or 0.1% of atropine. After 2 years of follow-up, all the atropine groups showed a significant effect on reducing myopia progression. In the 0.5% atropine group, around 60% of the children had shown a cessation in myopia progression, while 4% had a rapid progression [20]. Although 1% atropine was found to reduce myopia progression significantly, there were several intolerable side effects thus resulting in substantial dropouts, for which the major reason was photophobia [21].

The most frequently reported side effects of using topical atropine included light sensitivity and blurred near vision caused by pupil dilation and temporary paralysis of accommodation [22]. In the Atropine in the Treatment Of Myopia (ATOM) 1 study, no significant differences in the amplitude of accommodation or near visual acuity were found between atropine-treated and placebo-treated groups after they stopped using atropine within 6 months [23]. However, the reductions in accommodative amplitudes (11.3 D for atropine 0.01%, 3.8 D for atropine 0.1%, and 2.2 D for atropine 0.5%; *p* < 0.01) and pupillary enlargement (1 mm for atropine 0.01% vs. 3 mm for atropine 0.1% and 0.5%; *p* < 0.001) were shown in the ATOM 2 trial [24]. For all the three atropine administration groups with different doses, the accommodative amplitude was significantly less than that measured at the baseline visit at 36 months in the ATOM 2 study. There were no significant changes in intraocular pressure (IOP) between treatment groups, and no subjects with IOP greater than 21 mm Hg were reported after 2 years of treatment with 1% atropine [23]. In the multifocal electroretinogram (mfERG) study, the results were normal in the atropine group, and recordings of mfERG showed no significant effect on retinal function after two years of daily atropine usage [25]. On the contrary, ATOM 2 study has demonstrated a statistically significant reduction in the 30-Hz flicker and photopic A- and B-waves over time. However, multivariate analysis revealed that these changes correlated statistically more with axial elongation [26].

Therefore, this study aimed to address whether the use of atropine affects pupil diameter, IOP, and a variety of visual functions in children, as the cycloplegic is the most common treatment strategy in both the pre-myopic or early stage of myopia in Taiwan. The frequencies of the side effects and visual performance were also assessed.

## 2. Materials and Methods

### 2.1. Subjects

The approval from the Ethics Committee of Chung Shan Medical University Hospital (No: CS14095) was obtained and this study adhered to the Declaration of Helsinki. Both the parents and the children agreed to participate in the study and completed written informed consent. The subjects were recruited from two public elementary schools in Taichung city, the second-largest city by population in the middle of Taiwan, via vision screening activities. Two hundred and twelve children aged 6 to 12 years were selected randomly and enrolled in this study. They were divided into the atropine use group (who used atropine to control myopia successively at least 6 months prior to the examination; *n* = 99) and the control group (*n* = 113) based on their medication histories. Ophthalmologists prescribed the concentrations of atropine used by the subjects, who participated in the present study due to the policy of the public health administration, and the dose was recorded for each subject at the visit. Of the children in the atropine group, 55% used 0.125% daily eye drops and others with higher doses (the medications issued by the National Health Insurance Administration typically ranged from 0.125% to 0.5%).

The criteria of objective spherical equivalents ranging from +0.75 to −7.25 D were applied (including for the emmetropic and pre-myopic subjects). Subjective refraction was carried out for all the participants to obtain their best-corrected visual acuities (BCVAs) and two senior optometrists with over 5 years of clinical experience performed the test. Children with astigmatism >1.50 D, anisometropia >2.00 D, and those diagnosed with ocular injury or diseases such as amblyopia, squint, or cataract were excluded. None of the participants had a history of orthokeratology, corrective lenses wearing (including spectacle and contact lenses), and refractive/ocular surgery.

### 2.2. Procedures

All the participants completed a comprehensive examination, including physiological and functional evaluations and a questionnaire on the possible side effects of atropine use. The parameters of pupil size, intraocular pressure (IOP), and objective refraction were measured before visual function assessment. The pupil response was screened at the beginning for all the subjects to confirm drug administration, and their pupil sizes were measured using a pediatric autorefractor (binocular autorefractor, Plusoptix A09, Germany). The light entrance area of the pupil was calculated by using the formula of area of a circle: *A* = πr^2^ (*A*: area; r: radius of the horizontal pupil size) as round pupils were assumed for all the participants. The objective refractive error was measured using an open-field autorefractor (Shin-Nippon Nvision-K 5001, Osaka, Japan) before the assessment of visual function to minimize the influence of accommodative fluctuation. A noncontact air-puff tonometer (Topcon CT-80, Tokyo, Japan) was applied for IOP measurement.

The visual function assessment consisted of distance and near BCVAs, stereopsis, near point convergence, and accommodation amplitude. Distance visual acuities were measured with a 6-m Snellen chart using a projector, while a reduced Snellen acuity chart was used for near visual acuity test at 40 cm. The recordings were converted into LogMAR acuities for data analysis. The stereopsis of the children was determined using the Titmus stereo test and recorded as seconds of arc. The amplitude of accommodation was measured using the push-up method. Monocular estimated method retinoscopy (MEM) was also applied to confirm the difference in accommodative responses between the atropine group and their controls. A questionnaire on possible side effects that occurred during the period of atropine medications was completed by all the participants, and their daily visual complaints were also investigated.

### 2.3. Statistics

The statistical analysis of the present study was performed using SPSS software version 18.0 (SPSS, Inc, Chicago, IL, USA). The independent t-test was applied to examine the differences between the two groups, such as pupil size, IOP, and refractive error. The correlation between near visual acuity and pupil size was analyzed using Pearson’s correlation coefficients to test the relationship between variables. Data were presented as mean ± S.D., and *p* < 0.05 was considered statistically significant.

## 3. Results

### 3.1. Subjects

The control group consisted of 113 children who had never used atropine, and there were 99 children in the atropine group who had taken regular atropine medications for a consecutive 6 months. The characteristics of the two groups are shown in Table 1. There were no significant differences between the atropine group and their controls in age, height, and intra-ocular pressure. Highly significant differences (*p* < 0.001) were found in spherical equivalent, pupil size, light entrance area, and visual acuities between the two groups.

### 3.2. Effects of Atropine on Pupil Sizes, Intraocular Pressure, and Visual Acuity

The results showed that the horizontal pupil diameters of the right and left eyes of the atropine group were significantly larger than those of the control group. Therefore, Pearson correlations were applied to study the relationship between pupil size and visual acuity, the amplitude of accommodation, and the MEM measurements. The average of the results of both eyes was used for data analysis. There were no significant correlations between pupil size and any of the visual functions for the control group. However, a significant correlation of pupil size to near visual acuity (r = 0.254, *p* = 0.012) was found in the atropine group (Figure 1A), but not for distance visual acuity (r = 0.066, *p* = 0.47). There were also significant correlations between pupil diameter and accommodative responses (Figure 1B,C). No significant correlation was found between pupil size and IOP for both groups.

The children who used atropine were further classified as the low dose group (0.125%) and the higher dose group (0.3–0.5%). A statistically significant difference in pupil size was found between the two groups (*t* = −2.061, *p* = 0.042), which was consistent with the findings of the previous studies. The pupil size (mean ± SD) was 6.62 ± 1.04 mm for the low dose group, and 6.98 ± 0.85 mm for the higher dose group.

### 3.3. Effects of Atropine on Visual Function

Highly significant differences were found between the atropine group and the control group in the amplitude of accommodation, the results of near point convergence, and MEM findings (Table 2). There was a significant reduction in mean accommodative amplitude of the atropine group (OD: 7.41 ± 4.01 D, OS: 7.24 ± 3.83 D) compared to the control group (OD: 15.27 ± 5.24 D, OS: 15.82 ± 5.70 D), and the MEM results of the atropine group were significantly higher, suggesting that there was an accommodative lag in the atropine group. Both the means of NPC break and recovery results were significantly poorer in the atropine group (*p* < 0.001).

Regarding the stereopsis of the subjects, 177 subjects completed the Titmus stereo-test, and the results were collected for data analysis. 59.8% of the subjects in the control group had stereo-acuities smaller than 40 s of arc compared to 35.8% in the atropine group (Table 3). In addition, the proportion of the subjects who had poor stereopsis was higher in the atropine group than in the control (35.6% vs. 25%). Two hundred and five effective questionnaires were obtained on the side effects of atropine from both the atropine (*n* = 96) and the control group (*n* = 109). For all typical side effects of atropine administration, the atropine group showed higher frequencies of usually/always responses than the control group, particularly for photophobia and squinting (Figure 2). Around 74% of the children who used atropine reported that they had encountered photophobia, and 22.92% of the atropine group had diplopia, which was three times higher than the control group (7.34%). Regarding the quality of daily life vision, the subjects who had used atropine reported a complaint of blurred vision more frequently while reading, writing, and watching television, and more than 30% of the children reported that they had ever experienced a word or line skipping problem during the period of atropine treatment (Figure 3).

## 4. Discussion

Atropine is an antimuscarinic agent that is the most likely effective treatment for myopia control (dose-dependent). However, it is unfavored by patients due to its well-known side effect of photophobia. The use of cycloplegic eyedrops blocks both the pupillary sphincter and ciliary muscle, even though the mydriatic effect disappeared after drug cessation [27,28]. Therefore, it is important to think about the two major issues related to light sensitivity: dropout and UV-rays protection. In the study of Yen et al., 100% of the 1% atropine group reported the side effect of photophobia, and more than half of the subjects (151 of 247) dropped out from the clinical trial [19]. In Taiwan, Shih et al. also found that 22% of children assigned to 0.5% atropine experienced the problem of photophobia at the beginning of atropine treatment [20]. In the ATOM 1 study, 17% of the participants in the 1% atropine group dropped out for the study, and most had a severe light sensitivity problem [21,23]. The pupil was significantly dilated 12 h after instilling 0.01% atropine, and clinically significant short-term effects on pupil size lasted for at least 24 h [27]. From the data on the light entrance area calculated in the present study, the number of rays entering the eyes showed a 1.4-fold increase in the atropine group compared to the controls. Taiwan is a highly myopia prevalent country with possibly the longest history of atropine treatment for myopia control since the 1990s [19,20]. We have found a strong relationship between the severity of photophobia and patient compliance in the clinic, as full compliance was associated with decreased myopia progression during atropine treatment [29]. In the atropine group, it was also shown by the subjective questionnaire that 65.16% of the recruited children who were currently instilling the daily eyedrops replied that UV protection was not fully implemented outdoors. Even though they were informed to prevent light sensitivity by wearing sunglasses and hats, it is usually uncommon to use sun protection among the younger generation (6–18 years) in Asian countries. Therefore, potential retinal and lens phototoxicity effects should be a concern and monitored long-term by clinicians [10]. These limitations motivated the investigations into lower doses of atropine to reduce visual complaints while preserving the effect of myopia control.

Another issue accompanied by enlarged pupil diameter was the reduced visual acuity caused by an increase in all types of aberrations after instilling atropine [30,31]. A significant positive correlation between pupil diameter and near logMAR visual acuity was found in the atropine group (Figure 1A) but not for the control group. As described in the previous studies, reductions in the best-corrected visual acuities were found in the atropine group for both distance and near vision in the present study, suggesting that an increase of pupil diameter may play a role in decreasing image clarity. In the study of Wang et al., it has been demonstrated that all types of aberration increased significantly along with increasing pupil size (*p* < 0.001) with a more pronounced increase in spherical aberration coma aberration and high order astigmatism as pupil area increased [30]. Cycloplegia with cyclopentolate also significantly changed ocular higher-order aberrations (HOAs) from 0.124 ± 0.041 to 0.138 ± 0.037 μm for spherical-like aberrations, and from 0.279 ± 0.105 to +0.304 ± 0.096 μm for total HOAs, with no effects on coma-like aberrations and corneal HOAs after cycloplegia [31]. The reduced visual acuities of the atropine group observed in the present study compared to the controls suggested that the enlarged pupil size after mydriasis resulted in a degradation of optical images. A loss in near vision caused by cycloplegia was found in the Asian population-based trials [23,24] and shown in the studies of Caucasian children [32,33]. The near logMAR acuity was poorer in the eyes treated with 0.05% atropine compared to the 0.01% atropine group (0.05 ± 0.06 vs. −0.01 ± 0.06 logMAR), even though a significant difference was not found due to small sample number (*n* = 19) [32]. It has been demonstrated repeatedly that a combination of daily atropine treatment with distance and near correction, typically in the form of progressive addition lenses, can provide a synergistic effect on controlling the progression of myopia [34,35]. However, low dose atropine may be more appropriate to be implemented with ortho-keratology treatment to minimize the blurred near vision caused by accommodative lag [36].

The most relevant studies showed that high contrast best-corrected visual acuity at a distance was not affected in children during atropine treatment [37,38]. However, there was a significant difference between the atropine and control groups in the present study. In Taiwan, most parents choose topical atropine for myopia control because they refuse to use optical corrections at the onset of myopia. Higher frequencies of complaints of squinting and distance blur were also reported by the children who constantly used atropine to control the progression of myopia (see Figure 2 and Figure 3). There may be a few reasons for this finding: firstly, the doses used by the children in the present study (0.125–0.5%) were at least 10 times higher than the others, so a larger pupil size and more aberration may result in a worse acuity viewing at a distance. Atropine 1% initially produced a linear reduction of about 0.2 logMAR lines per diopters in distance acuity in uncorrected hyperopes [39]. Secondly, in Taiwan, atropine has been regarded as a myopia treatment to substitute or avoid wearing eyeglasses for the parents, particularly at the early stage of myopia development. Thus, the chance of obtaining a clear vision would be sacrificed, and the children undergo a period of blur vision without being corrected. It was found that among 18 children who were diagnosed with myopic spherical anisometropia, 13 patients (72.2%) had developed amblyopia [40]. The lack of exposure to a clear visual environment might cause a reduction in the mean BCVA of the atropine group in the study. However, this phenomenon of pseudo-amblyopia disappeared after applying their first optical corrections, usually when arresting an average of myopia of 2.00 D to 2.50 D. Thirdly, the subjective refraction and BCVA measurements were conducted at the elementary schools for larger sample size collections. The values of the BCVAs could be underestimated compared to the data collected in standard examination rooms.

Just as in the evidence shown in the previous studies, accommodation was significantly affected by daily atropine eyedrops [27,39]. In the ATOM 2 trial, accommodation was less in the 0.5% atropine group (13.24 ± 2.72 D) compared with the 0.1% atropine (14.45 ± 2.61 D) and 0.01% atropine group (14.01 ± 2.9 D; *p* < 0.001) [24]. The authors suggested that the diminished accommodative amplitude after the 1-year washout might become a permanent loss of accommodative ability, and the amount increased with higher doses of atropine. The ATOM 2 study reported that the mean accommodative amplitude reduced by approximately 4.50 D over the two years of treatment in the 0.01% atropine group, and Cooper et al. suggested that 0.02% was the highest concentration that resulted in a clinically acceptable level of pupillary dilation and loss of accommodation without subjective symptoms [33]. Low-concentration atropine (0.01–0.05%) has also shown no significant effect on the amplitude of accommodation across all the groups in the LAMP study [41]. It was also noted that in the present study, the reduction in amplitude of accommodation and more accommodative lag observed in the atropine group accompanied more subjective complaints of reading/writing blur, words or lines skipping, diplopia, and significantly longer break and recovery points. In addition, only 35.3% of the atropine group had standard stereopsis (≦40 s of arc) compared with 59.8% in the control group as the stereo acuities were measured with full correction. All concentrations of atropine could result in side effects. Therefore, it is essential for clinicians to judge the efficacy of myopia control, visual function as well as rebound effect after drug cessation for each individual child.

## 5. Study Limitations

(i)Due to the issue of drug administration, the information of subject medication history can only be collected by using a questionnaire and which mainly relies on the parents’ memory. So, it would be optimal to collaborate with ophthalmologists in the future for both population allocation and medical database sharing.(ii)The concentrations of the atropine eyedrops were recorded based on the medications the children were using at the time point of data collection. However, there may be some who changed the doses in the past 6 months prior to the examination. Again, it would be beneficial to work with ophthalmologists to solve this problem and keep the doses fixed.(iii)To examine the effect of atropine dose on visual function and ocular physical changes, it is required to recruit more participants using different doses of atropine eyedrops.

## 6. Conclusions

The results of enlarged pupil size, reduced visual acuity, and decreased accommodative amplitude after instilling topical atropine were consistent with the previous findings. Greater ratios of abnormal stereopsis and subjective visual complaints accompanied the use of atropine. Long-term myopia management with a comprehensive eye examination is required, including monitoring children’s visual acuities and minimizing their visual discomforts. More large-scale longitudinal studies and/or retrospective studies among young adults should be conducted to provide evidence of a long term effect of topical atropine on controlling myopia progression and preventing high myopia.

## Figures and Tables

**Figure 1 children-08-01054-f001:**
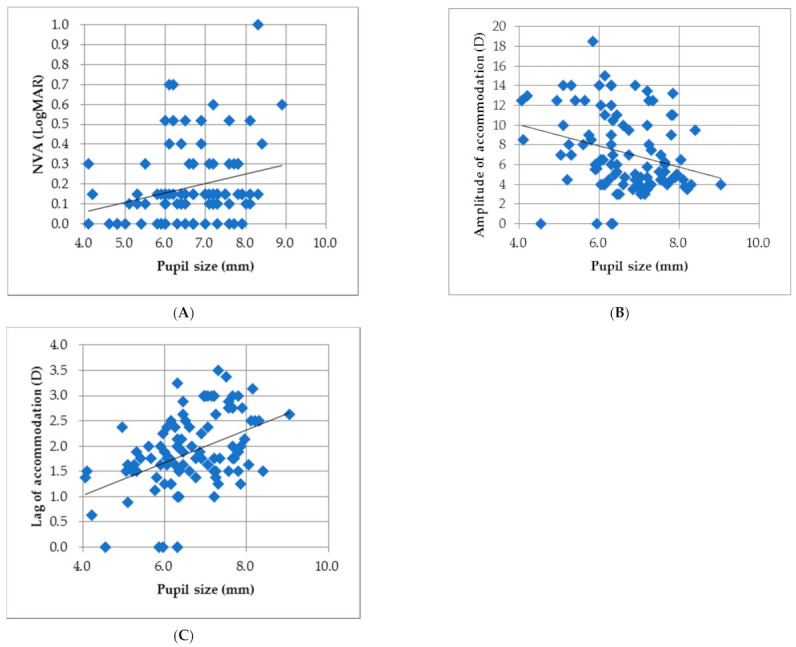
Statistically significant correlations were found between (**A**) pupil size and near LogMAR acuity (r = 0.254, *p* = 0.012); (**B**) pupil size and amplitude of accommodation (r = −0.275, *p* = 0.006); (**C**) pupil size and accommodative lag (r = 0.428, *p* < 0.0001) in the atropine group.

**Figure 2 children-08-01054-f002:**
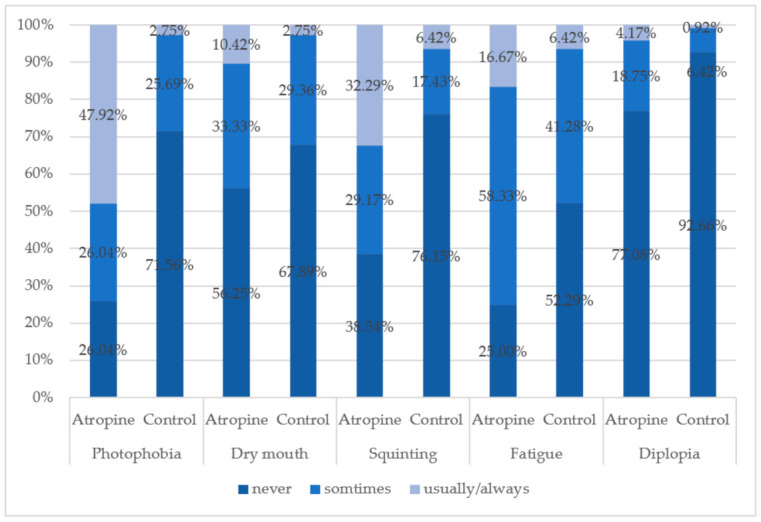
The frequencies of the five major atropine administration-induced side effects between the atropine group and the control group.

**Figure 3 children-08-01054-f003:**
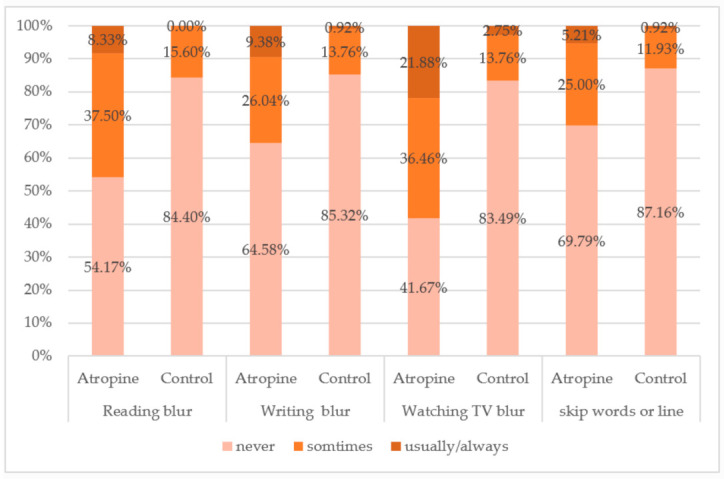
The frequencies of the four chief complaints regarding daily visual performance between the atropine and control groups.

**Table 1 children-08-01054-t001:** Comparison of characteristics between the atropine group and the control group.

		Control Group	Atropine Group	*p*-Value
Number		113	99	
Age (yr)		10.19 ± 1.21	10.38 ± 1.16	0.246
Hight (cm)		139.77 ± 9.25	139.26 ± 9.48	0.699
IOP (mmHg)	OD	15.84 ± 3.61	15.99 ± 3.47	0.765
	OS	15.63 ± 3.21	15.87 ± 3.06	0.598
Spherical equivalent (D)	OD	−0.59 ± 1.41	−1.96 ± 1.46	<0.001 **
	OS	−0.52 ± 1.36	−2.04 ± 1.51	0.001 **
Pupil size (mm)	OD	5.40 ± 0.90	6.60 ± 1.01	<0.001 **
	OS	5.42 ± 0.87	6.64 ± 1.00	<0.001 **
Light entrance area (mm^2^)	OD	20.63 ± 7.74	28.95 ± 11.11	<0.001 **
	OS	20.67 ± 7.41	29.20 ± 10.92	<0.001 **
Distance visual acuity (logMAR)	OD	0.12 ± 0.28	0.45 ± 0.35	<0.01 *
	OS	0.11 ± 0.28	0.50 ± 0.34	<0.01 *
	OU	0.05 ± 0.25	0.36 ± 0.32	<0.01 *
Near visualacuity (logMAR)	OD	0.06 ± 0.08	0.19 ± 0.19	<0.001 **
	OS	0.06 ± 0.08	0.167 ± 0.17	<0.001 **
	OU	0.04 ± 0.07	0.12 ± 0.16	<0.001 **

The values shown here were the corrected visual acuities. Abbreviations: OD: right eye; OS: left eye; OU: both eyes; IOP: intraocular pressure. Statistical significance: * *p* < 0.01; ** *p* < 0.001.

**Table 2 children-08-01054-t002:** The comparison of accommodative responses and near point convergence between the atropine group and the control group.

	Control Group	Atropine Group	*p*-Value
AA_OD (D)	15.27 ± 5.24	7.41 ± 4.01	<0.001 **
AA_OS (D)	15.82 ± 5.70	7.24 ± 3.83	<0.001 **
NPC_break point (cm)	4.23 ± 3.63	12.82 ± 8.19	<0.001 **
NPC_recovery point (cm)	6.37 ± 4.29	15.27 ± 8.60	<0.001 **
MEM_OD (D)	0.86 ± 0.52	1.89 ± 0.73	<0.001 **
MEM_OS (D)	0.95 ± 0.48	1.94 ± 0.71	<0.001 **

Statistical significance: ** *p* < 0.001.

**Table 3 children-08-01054-t003:** Comparison of the stereo acuities between the atropine group and the control group.

	Control Group		Atropine Group	
	*n*	%	*n*	%
	92		85	
Good (≤40)	55	59.8	30	35.3
Average (41–63)	14	16.2	23	27.1
Poor (64–100)	15	16.3	20	21.5
Very poor (>100)	8	8.7	12	14.1

## Data Availability

The raw data supporting the conclusions of this article will be made available by the authors.

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
