# Peer review of "The Impact of Clinical Atropine Use in Taiwanese Schoolchildren: Changes in Physiological Characteristics and Visual Functions"

_children, 2021, doi:10.3390/children8111054_

Round 1

Reviewer 1 Report

Dear Authors,

thank you very much for this interesting paper. 

please clarify what kind of study you performed clearly in the abstract and methods section (prospective/ retrospective, controlled, randomized?/observational). From the current description it is not entirely clear.

Who decided what child would get atropin and what child would not?

Also, please include a paragraph about the limitations of the study.

Kind regards!

Author Response

Responses:

Thank you very much for the helpful comments!

Point 1-

The design of the present study was a cross-sectional study investigating the influence of atropine use in Taiwanese children. The subjects were selected from the elementary schools based on their medication histories, so it is a kind of controlled and randomized study design despite the children were actually grouped by ophthalmologists according to whether they received the prescription of atropine or not. Please see the changes in the Abstract and Methods sections (line 107-108).

Point 2-

The ocular medications can only be determined by ophthalmologists in Taiwan, and therefore, we cannot assign the participants by ourselves. However, the children who needed atropine treatment were mostly myopic or close to being myopic, and their parents accepted it as a therapy without optical corrections at the early stage of myopia progression. Please see the interpretations in the manuscript (line 113-118).

Point 3-

Please find the inserted paragraph for the study limitations (line 327-339).

Reviewer 2 Report

  1. Please include the discussion on low dose atropine (0.01-0.05%) use for slowing myopia progression (LAMP study). Why all the children were prescribed higher doses ? Low doses seem to work as well as higher doses with fewer side effects.
  2. Was objective refraction error measured after cycloplegia ? It is the only sensible way to assess refraction error in children. If so, please place that information in the text.
  3. For me, a very interesting subject is the difference in visual function between low dose and high dose atropine. Is it possible to provide such comparison ? Result would be of great practical value. Please make that effort and provide additional analysis.

Author Response

Responses:

Point 1- The findings of the LAMP study have been added in the section of Discussion. Please see the change in the manuscript (line 318-320). The ocular medications can only be determined by ophthalmologists in Taiwan. Please see the interpretations in the manuscript (line 113-118).

Point 2- Due to the same issue as the above response, cycloplegic refraction was not allowed during data collection. Instead, an open-field autorefractor with a distant target was applied for measuring the children’s refractions to relax their accommodation (line 136-139).

Point 3- The differences in visual function were found between the low and higher concentration group. However, the results will be shown in another paper. Here are the relevant information from the statistical reports:  

0.125 % (n=54)

> 0.2 % (n=45)

P value

Mean ± SD

Pupil size

6.16±1.16

6.57±1.08

0.06†

Near VA

0.058±0.075

0.127±0.145

0.001

Reviewer 3 Report

This is an interesting paper reporting on the side effects of atropine use.

To me, it is not quite clear whether best-corrected visual acuity was assessed or not. In the "Methods" section, it is mentioned that BCVA was assessed, in the results, however, it seems more like only visual acuity without correction is reported. I think BCVA would also be of interest to the readers.

Author Response

Responses:

Thank you very much for the comments!

Due to the limitations of insufficient examination time and the unfavorable site which were located inside the schools (including lighting, traffic congestion), it would be difficult to perform a thorough subjective refraction exam. For future study, it is required to invite children to come to the examination room for a comprehensive examination. Please see the interpretations in the manuscript (line 292-307).

However, for the distance BCVA, we observed that most of the children in the atropine group spent a long time on completing subjective refraction and a significant reduction in their VAs. This could be because that the larger pupil size leads to more optical aberration, but the possibility of pseudo-amblyopia can’t be totally ruled out.

Round 2

Reviewer 1 Report

Dear Authors,

thank you for making the required changes.

KInd regards!

Author Response

Dear Reviewer,

We appreciated all the efforts that you provided for improving the manuscript!

Best regards,

Reviewer 3 Report

The authors provided a revised version of the manuscript.

In this version, however, the authors state that the study was randomized, how was this done? To my understanding, the atropine group already has used atropine while the control group mainly was not myopic. Please clarify.

I also noted when going through the manuscript that the axes of the figures are in chinese language.

In Table 1, it should be stated that BCVA was assessed, as the authors confirmed in their response to the reviewers.

Author Response

Dear Reviewer, 

Thank you very much for reminding us the important thing that the basis of the two groups was different (close to be myopic vs. myopic), and thus influencing their medical therapies, even though their medications were determined by ophthalmologists rather than the study itself. (Please see the changes in line 107-108)

The notation of the corrected visual acuities has been added in Table.1. However, because we cannot guarantee that it's the "Best" corrected VA, it would be more appropriate to avoid using the term "BCVA". (Please see the changes in line 171)

The figures presented in my computer were normal, but a PDF file was also attached just in case if there is any error of the figures. 

Again, we appreciated for all the efforts that you made for the manuscript!

Kind regards,